# *OsJRL40*, a Jacalin-Related Lectin Gene, Promotes Salt Stress Tolerance in Rice

**DOI:** 10.3390/ijms24087441

**Published:** 2023-04-18

**Authors:** Qinmei Gao, Xiaolin Yin, Feng Wang, Shuchang Hu, Weihao Liu, Liangbi Chen, Xiaojun Dai, Manzhong Liang

**Affiliations:** Hunan Province Key Laboratory of Crop Sterile Germplasm Resource Innovation and Application, Hunan Normal University, Changsha 410081, China; 15802620355@163.com (Q.G.); yinxleffort@163.com (X.Y.); 13975877142@163.com (F.W.); hsc19990221@163.com (S.H.); 18749958161@163.com (W.L.); chenliangbi@126.com (L.C.)

**Keywords:** rice (*Oryza sativa* L.), *jacalin*, *OsJRL40*, salt stress, Na^+^/K^+^ ion

## Abstract

High salinity is a major stress factor affecting the quality and productivity of rice (*Oryza sativa* L.). Although numerous salt tolerance-related genes have been identified in rice, their molecular mechanisms remain unknown. Here, we report that *OsJRL40*, a jacalin-related lectin gene, confers remarkable salt tolerance in rice. The loss of function of *OsJRL40* increased sensitivity to salt stress in rice, whereas its overexpression enhanced salt tolerance at the seedling stage and during reproductive growth. β-glucuronidase (GUS) reporter assays indicated that *OsJRL40* is expressed to higher levels in roots and internodes than in other tissues, and subcellular localization analysis revealed that the OsJRL40 protein localizes to the cytoplasm. Further molecular analyses showed that OsJRL40 enhances antioxidant enzyme activities and regulates Na^+^-K^+^ homeostasis under salt stress. RNA-seq analysis revealed that *OsJRL40* regulates salt tolerance in rice by controlling the expression of genes encoding Na^+^/K^+^ transporters, salt-responsive transcription factors, and other salt response-related proteins. Overall, this study provides a scientific basis for an in-depth investigation of the salt tolerance mechanism in rice and could guide the breeding of salt-tolerant rice cultivars.

## 1. Introduction

Rice (*Oryza sativa* L.) is the main agricultural crop in the world and the staple food of most of the global population [1]. Salt stress is one of the important factors affecting rice production. Approximately 7% of the total land area is affected by high salinity [2,3]. Excess salt adversely affects many plant processes including photosynthesis, metabolism, and respiration, resulting in stunted growth and even death [4]. High salt can also cause Na^+^ poisoning, which seriously affects plant growth and development and consequently reduces grain quality and yield in rice. Plants respond to the unfavorable environment surrounding them by generating complex inter-gene interactions and cross-talk with different molecular pathways [5]. How plants adapt to salt stress and how their salt tolerance levels can be improved further have long been the hot topics of research in the field of plant physiology and ecology.

Plants have evolved several mechanisms of salt tolerance, including membrane modification, ion absorption, transport, and separation, and salt excretion, etc. [6]. Many genes and gene families have been reported to play important roles in plant salt tolerance. For example, *OsZFP213* and *OsMADS25* are involved in regulating the maintenance of the antioxidant system, which improves the salt tolerance of rice by reducing the electrical conductivity and the accumulation of harmful substances such as hydrogen peroxide and malondialdehyde (MDA) [7,8]. High-affinity K^+^ transporter (KHT) [9,10], Na^+^/H^+^ antiporter (NHX) [11,12], and high-affinity K^+^ (HAK) transporter [13] gene families regulate Na^+^-K^+^ balance under salt stress, thus promoting salt tolerance. In addition, plant hormones are involved in plant response to salt stress, among which abscisic acid (ABA) plays a key role in regulating plant salt tolerance by regulating the expression of salt-stress-related genes [14,15]. Recent studies showed that lectin-related genes play an important role in stress tolerance. Lectins are stress-protective proteins that specifically and reversibly bind to carbohydrates, as shown in rice [16,17,18], localize to the cell cytoplasm or nucleus, and interact with various stress factors and storage proteins or enzymes during plant development [16,18]. Studies show that lectin-related genes are expressed under abiotic and biotic stresses in various plant species including wheat (*Triticum aestivum*) [19], *Arabidopsis thaliana* [20], soybean (*Glycine max*) [21], and rice [16], indicating that these genes are involved in biotic and abiotic stress response. Transcriptome analysis found that some lectin genes exhibit different expression levels under drought and salt stresses [17]. Lectins are not only involved in stress protection but also in plant development, communication, intercellular communication [22], and inhibition of insect and virus growth [23]. Lectins are divided into different families: jacalin-related lectin family, legume lectin family, and LysM domain lectin family.

Jacalin-related lectin (*JRL*) genes have attracted much attention in recent years [24]. The JRL family is the third largest lectin family in rice. Jacalin lectin is a sugar-binding domain mainly found in plants. The jacalin domain of lectin is related to its function, and other domains bound to jacalin participate in a variety of biotic and abiotic stress responses [25]. Functionally, most JRLs are related to disease resistance and multiple stress response signals [26,27]. Nearly half of the lectin genes show differential expression under biotic and/or abiotic stresses [16]. Studies show that JRLs are composed only of jacalin and are stress proteins related to plant disease, insect resistance, injury response, and abiotic stress response [17]. OsJAC1 is a mannose-binding JRL derived from rice (Oryza sativa) [28], which regulates growth development [29] and is involved in plant defense against pathogen attacks [30]. The expression of HvHorcH enhanced its salt tolerance in Arabidopsis thaliana root tip [31]. In rice, *Orysata* is up-regulated under various stresses (salt, senescence, pathogen attack, insect attack, hormone, etc.) [32,33,34,35]. Early studies by Zhang and colleagues showed that salt induces the expression of *JRL* genes in rice [36]. He et al. found that overexpression of *OsJRL* increased the sensitivity of rice to ABA and affected the expression of some salt stress-related genes [33]. In recent years, similar results have been obtained in several additional plant species including wheat [16], poplar (*Populus euphratica*) [37], and barley (*Hordeum vulgare*) [31], suggesting that the *JRL* family genes play an important role in the salt stress response. In recent years, JRLs have become an important focus of research in the field of plant biochemistry, physiology, and medicine; however, there are few studies on the salt tolerance of JRLs in rice.

In our previous work on the identification of salt tolerance-related candidate genes in ‘Sea Rice 86′(SR86) using quantitative trait locus (QTL)-Seq and bulked segregant analysis (BSA)-Seq, we found a JRL family gene may be an important candidate gene and is named *OsJRL40*; RT-qPCR analysis showed that this gene actively responds to salt stress [38], but its molecular function was unclear. In the present study, we investigated the spatiotemporal expression patterns of *OsJRL40*, subcellular localization of the encoded protein, and salt tolerance phenotype of *OsJRL40*-overexpressing transgenic plants under salt stress. The antioxidant and ion homeostasis capacities of OsJRL40 were also analyzed to clarify its physiological and molecular mechanisms of action. Additionally, transcriptomic analysis of wild-type (WT) and *osjrl40* knockout mutant and *OsJRL40*-overexpressing plants was performed under normal and salt stress conditions. The results showed that *OsJRL40* regulates salt tolerance in rice by regulating Na^+^-K^+^ transport-related genes, transcription factors associated with salt response, and other salt response-related genes.

## 2. Results

### 2.1. Analysis of OsJRL40 Amino Acid Sequence and Phylogenetic Relationship

*Os04g0123600*, which is located on chromosome 4, has a coding sequence (CDS) of 426 bp and encodes 141-amino acid JRL protein. Amino acid sequence alignment analysis showed that Os04g0123600, barley mJRL protein LEM2, TaJRLL1 [37], AtJAL33 [39], and AtNSP4 [40] possess nine invariant non-polar amino acid residues in the jacalin lectin domain, which are essential for protein folding and mannose-binding [41,42]. A similar mannose-binding site in Heltuba protein has a highly conserved region near the C-terminal end [41] (Figure 1A). These results suggest that the *Os04g0123600* encodes an mJRL protein, named OsJRL40.

Next, we conducted the phylogenetic analysis of JRL proteins of rice, maize, wheat, Arabidopsis, and soybean, whose protein sequences are derived from Song et al. [19]. OsJRL40 was found to be most closely related to maize (Figure 1B).

### 2.2. Expression and Subcellular Localization Assay of OsJRL40

To verify the function of *OsJRL40*, real-time fluorescence quantitative PCR (RT-qPCR) analysis was performed. The expression of *OsJRL40* was most strongly induced when treated with 100 mM NaCl, reaching a peak at 6 h and then declining, indicating that *OsJRL40* is a salt stress-responsive gene (Figure 2A). *OsJRL40* was also induced in response to exogenous abscisic acid (ABA) application (Figure 2B). Next, we analyzed the *OsJRL40* expression pattern in different tissues of *japonica* rice Nip by RT-qPCR. *OsJRL40* showed a higher expression level in the stem than in other tissues (Figure 2C). To further examine the expression pattern of *OsJRL40* in different tissues of rice plants, a 2.5-kb *OsJRL40* promoter fragment driving the *β-glucuronidase* (*GUS*) reporter gene was introduced into the rice. GUS staining was detected in roots, stems, leaves, internodes, bud sheaths, and panicles. The cross-sections of roots and leaves revealed staining of the xylem and vascular systems, respectively (Figure 2D). 

To investigate the subcellular localization of OsJRL40, the *OsJRL40-GFP* fusion was transiently expressed in Nip protoplasts under the control of the cauliflower mosaic virus *35S* (CaMV35S) promoter. Green fluorescence was observed in the cytoplasm (Figure 2E), indicating that OsJRL40 is a cytosolic protein.

### 2.3. OsJRL40 Positively Regulates Salt Stress Tolerance in Rice at the Seedling Stage

In a previous study, we preliminarily determined that *OsJRL40* is involved in regulating salt tolerance in rice [38]. To investigate the possible function of *OsJRL40* in the salt stress response, two *OsJRL40* loss-of-function knockout mutant lines (KO1 and KO2) were generated in the Nip background using the CRISPR/Cas9 technology. The KO1 mutant contained a 1-nt deletion in the first exon of *OsJRL40*, causing premature termination of protein, while the KO2 mutant contained a 1-nt insertion in the second exon of *OsJRL40*, resulting in frameshift mutation (Figure 3A). We also generated two independent *OsJRL40* overexpression lines (OE1 and OE2) in the Nip background. RT-qPCR indicated that *OsJRL40* was significantly up-regulated in both OE1 and OE2 compared with the WT (Nip) (Figure 3B). Next, we evaluated the salt tolerance of WT, KO, and OE plants at the seedling stage. The KO1 and KO2 seedlings showed severe wilting, while the OE1 and OE2 seedlings grew better than the WT (Figure 3C). Compared with the WT, the survival rate, fresh weight, and chlorophyll content of KO1 and KO2 lines were significantly lower, while those of OE1 and OE2 were higher after the NaCl treatment (Figure 3D–F). These results indicate that *OsJRL40* positively regulates salt stress tolerance in rice.

### 2.4. OsJRL40 Enhances Salt Tolerance in Rice at the Reproductive Stage

To determine whether *OsJRL40* affects salt tolerance in rice at the reproductive stage, we examined the phenotypes and agronomic traits of WT, KO, and OE plants under normal (0 mM NaCl) and salt stress (150 mM NaCl) conditions (Figure 4). Under normal conditions, the thousand-grain weight of KO plants was lower, while that of OE plants was higher than the WT (Figure 4H), and no significant differences were observed in the other traits (Figure 4). Under salt stress conditions, all KO plants died, while the OE lines showed significantly greater plant height, panicle length, seed setting rate, and thousand-grain weight than the WT (Figure 4D,F–H). The WT, KO, and OE plants showed no significant difference in panicle number under salt stress conditions (Figure 4E). These results suggest that *OsJRL40* enhances the salt tolerance and yield of reproductive-stage rice plants under salt stress. 

### 2.5. ABA Affects Seed Germination and Plant Growth of OE Lines

According to previous research [43], overexpression of *JRL* family genes increases the sensitivity of plants to ABA and alters the expression of downstream genes. RT-qPCR analysis revealed that the *OsJRL40* was responsive to exogenous ABA (Figure 2B). Under normal conditions, the germination rate of KO and OE seeds was similar to that of WT seeds. However, in the 2 μM ABA treatment, the germination rate of KO seeds was higher than that of WT seeds, whereas the germination rate of OE seeds was lower than that of WT seeds. In the 5 μM ABA treatment, the seed germination rate of KO lines was approximately 30%, whereas that of OE seeds was almost 0 (Figure 5A,B).

To further investigate the relationship between *OsJRL40* expression and exogenous ABA application, we germinated the WT, OE, and KO seeds on half-strength Murashige and Skoog (1/2 MS) medium for 2 days and then transferred them to 1/2 MS medium containing varying concentrations of ABA to evaluate seedling growth. On normal medium, the growth of the KO and OE lines was similar to that of WT plants (Figure 5C–E). However, in the presence of 2 and 5 μM ABA, the roots and shoots of WT plants were shorter than those of KO lines and longer than those of OE lines (Figure 5C–E). These results indicate that overexpression of *OsJRL40* increases the ABA sensitivity of rice plants, whereas knockout mutation of *OsJRL40* has the opposite effect. 

### 2.6. OsJRL40 Enhances the Antioxidant Capacity of Rice under Salt Stress

To explore the molecular mechanism of OsJRL40-induced salt stress tolerance in rice, we studied the physiological and biochemical parameters of WT, KO, and OE plants under salt stress. No significant differences were observed among the three genotypes under normal conditions. However, a 2 d treatment with 100 mM NaCl led to the accumulation of reactive oxygen species (ROS) as revealed by diaminobenzidine (DAB) staining, which was stronger in KO mutants than in the WT (Figure 6A,B). Additionally, the content of hydrogen peroxide (H_2_O_2_) was higher in KO mutant leaves than in WT leaves (Figure 6D). The malondialdehyde (MDA) content and relative electrical conductivity (EC) were significantly higher in KO mutants (Figure 6C,E), which is consistent with their performance under salt stress. The trends displayed by OE plants, in terms of ROS and MDA contents and EC, were opposite to those displayed by KO plants. The activities of ROS scavenging enzymes are closely related to the salt stress tolerance levels of plants [44]. Consistently, the activities of superoxide dismutase (SOD) and peroxidase (POD) were dramatically lower in KO1 and KO2 lines and significantly higher in OE1 and OE2 lines than in the WT (Figure 6F,G). These results indicate that OsJRL40 improves antioxidant enzyme activity in rice under salt stress. 

### 2.7. OsJRL40 Regulates Na^+^-K^+^ Homeostasis under Salt Stress

To test whether the growth phenotype of KO1 and KO2 lines under high salinity conditions was caused by the overaccumulation of Na^+^, we determined the Na^+^ and K^+^ contents of the roots and shoots of WT, KO1, KO2, OE1, and OE2 plants. No differences in Na^+^ and K^+^ contents were detected among the different genotypes under normal conditions. However, under salt stress conditions, the KO1 and KO2 plants accumulated more Na^+^ (Figure 7A,D), but less K^+^ (Figure 7B,E), in shoots and roots than the WT plants, while the OE plants showed the opposite results. The Na^+^/K^+^ ratio was significantly higher in the shoots and roots of KO1 and KO2 plants than in those of WT, OE1, and OE2 plants (Figure 7C,F). Taken together, these results suggest that OsJRL40 enhances salt tolerance in rice by reducing Na^+^ accumulation in the shoot and root, thus modifying the Na^+^/K^+^ ratio. 

### 2.8. OsJRL40 Mediates Salt Tolerance in Yeast

To further clarify the function of *OsJRL40*, we performed functional assays in yeast (*Saccharomyces cerevisiae*) strain BY4741. The plasmid containing *OsJRL40* was transformed into BY4741 cells, and the transformed cells were grown on media containing different concentrations of NaCl (Figure 8). Under normal conditions (0% NaCl), no significant difference was detected between yeast transformants carrying the empty pYES2 vector (control) and those expressing *OsJRL40* (Figure 8). Under salt stress conditions, the growth of BY4741 cells carrying the empty vector was inhibited more significantly than that of yeast cells expressing *OsJRL40*, and the growth inhibition of yeast cells increased with the increase in NaCl concentration (Figure 8). These results suggest that *OsJRL40* enhances the salt tolerance of yeast. 

### 2.9. OsJRL40 Affects the Expression Profiles of Stress-Responsive Genes

To further clarify the mechanism of action of *OsJRL40* at the molecular level, we performed RNA-seq analysis on WT and KO1 plants grown under normal (0 mM NaCl) or salt stress (100 mM NaCl) conditions. A total of 2533 genes showing differential expression between WT and KO1 plants were identified, of which 446 were up-regulated and 1406 were down-regulated in KO1 plants compared with the WT (Figure 9A). Under normal conditions, 254 genes were up-regulated and 229 genes were down-regulated in KO1 plants compared with the WT (Figure 10A). Kyoto Encyclopedia of Genes and Genomes (KEGG) enrichment analysis showed that these differentially expressed genes (DEGs) were involved in plant-pathogen interaction, sesquiterpenoid and triterpenoid biosynthesis, and mitogen-activated protein kinase (MAPK) signaling pathway (Figure 9B), and Gene Ontology (GO) enrichment analysis showed that these DEGs were involved in defense response, DNA integration, and ADP binding (Figure 9C). Under salt stress conditions, 192 genes were up-regulated and 1177 genes were down-regulated in KO1 plants compared with the WT (Figure 10B). KEGG enrichment analysis showed that these DEGs were enriched in plant-pathogen interaction, flavonoid biosynthesis, phenylpropanoid biosynthesis, and MAPK signaling pathway (Figure 10C). GO enrichment analysis showed that most of these DEGs were involved in response to wounding, defense response, stress regulation, and so on (Figure 10D). 

Because most of the DEGs were down-regulated after salt treatment and most of them were involved in stress response, we focused on these down-regulated genes. A comparison of the DEGs identified in KO plants before and after salt stress treatment revealed that 1021 DEGs were only down-regulated in KO plants. These down-regulated genes were defined as the candidate targets of *OsJRL40* (Figure 11A) and were further annotated by KEGG and GO enrichment analyses. The results of KEGG enrichment analysis showed that most of these DEGs were involved in plant-pathogen interaction, hormone signal transduction, MAPK signaling pathway, and the biosynthesis and metabolism of phenylpropanoids, starch and sucrose, and flavonoids. The results of GO enrichment analysis showed that most of the genes were involved in stress response, stress regulation, and bind activity, and some genes were involved in the salt stress response regulation pathway (Figure 11B,C). We focused on some genes involved in stress response pathways, including transcription factor genes (*WARKY54, WARKY71, MYB2,* and *NAC19*), Na^+^-K^+^ ion transporter genes (*OsHAK1, OsHAK17, OsKAT1, OsAKT1*, and *OsAKT2*), and some salt stress response-related genes. Most of these genes have been reported to respond to salt stress and play an important role in salt tolerance. The OsHAK1 regulates salt tolerance in rice by regulating K^+^ concentration [13]. The OsKAT1 confers salinity tolerance to yeast and rice cells [45]. The expressing *NAC19* gene increased the salt tolerance of plants, and the leaf water content and chlorophyll content of these plants were higher than those of the WT [46]. OsWRKY54 regulates the expression of some essential genes related to salt tolerance [47]. The expression of the *OsWRKY71* transcript level was the highest under salinity stress [48]. OsGAP1 attenuates salt stress and binds to phospholipids and the unconventional G protein OsYCHF1 [49].

To confirm the results of RNA-seq analysis, we selected eight genes for RT-qPCR analysis. Our RT-qPCR results indicated that these genes were up-regulated in WT plants and down-regulated in KO plants under salt stress (Figure 12A), which was consistent with our RNA-seq results. Collectively, RNA-seq analysis revealed that OsJRL40 regulates salt tolerance in rice by regulating Na^+^-K^+^ transport-related genes, salt response-associated transcription factor genes, and other salt response-related genes.

## 3. Discussion

The JRL family is the third largest family of lectins in rice. Transcriptome and whole-genome sequencing analyses show that *JRL* family genes are involved in the salt stress response, although the experimental evidence available is limited. Here, we provide genetic evidence demonstrating that the *JRL* family gene *OsJRL40* positively regulates tolerance to salt stress in rice. *OsJRL40* was mainly expressed in nodes and responded quickly to salt stress. The *osjrl40* KO mutants were sensitive to salt stress, while *OsJRL40* OE lines exhibited enhanced salt stress tolerance. In addition, the OE plants showed strong salt tolerance at the reproductive stage, as evident from their increased panicle length, seed setting rate, thousand-grain weight, and consequent yield under salt stress conditions. Under salt stress conditions, *OsJRL40* overexpression enhanced the antioxidant capacity of cells and eliminated excessive ROS accumulation. In addition, OsJRL40 regulated the Na^+^-K^+^ balance and the expression of many stress-related genes, suggesting that *OsJRL40* plays an important role in salt stress tolerance. 

In most plant species, salt stress tolerance is associated with a decrease in Na^+^ accumulation in aerial plant parts. Plants have evolved various mechanisms to enhance salt tolerance, including increasing the Na^+^-chelating capacity of vacuoles and promoting the transfer of Na^+^ from bud to root via the phloem sap [50]. According to previous studies, K^+^ channel proteins including HAKs, AKTs, and KATs play important roles in salt stress tolerance in rice [51,52]. *HAK* family genes regulate K^+^ uptake and translocation, salt tolerance, and osmotic potential regulation [13,53]. The expression of *OsAKTs* and *OsKATs* is induced in the root and stem tissues of rice in response to salt stress. The K^+^ transporter OsHAK1 regulates salt tolerance in rice by regulating K^+^ concentration [48]. *AKT1*, *AKT2*, and *KAT1* are induced by salt stress [54]. The rice Shaker K^+^ channel protein OsKAT1 confers salinity tolerance to yeast and rice cells [45]. While the loss-of-function mutation of *OsJRL40* increased the Na^+^ content of shoots and roots, its overexpression had the opposite effect. This suggests that OsJRL40 regulates K^+^ transporters to maintain Na^+^-K^+^ homeostasis in rice under salt stress conditions. RT-qPCR results showed that, under salt stress conditions, the expression levels of *OsHAK1*, *OsAKT1*, and *OsKAT1* were significantly lower in KO mutant plants and significantly higher in OE plants than in the WT. These results will serve as a reference for in-depth research on the salt tolerance mechanism of *JRL* family genes. 

RNA-seq analysis showed that 1406 and 162 genes were down-regulated in the KO mutant and OE plants, respectively, compared with the WT. KEGG enrichment analysis demonstrated that most DEGs were involved in plant hormone signaling and MAPK signaling pathways and substance anabolic pathways. GO functional analysis showed that most DEGs were involved in stress response regulation. Our results showed that *OsJRL40* was also responsive to exogenous ABA application, and the OE lines were sensitive to ABA. Previous research shows that members of the OsJRL family regulate ABA response and the expression of genes involved in the ABA signaling pathway [43]. Poplar *JRL* genes mediate the response to ABA under salt stress, maintaining ROS and ion homeostasis [37]. Thus, it is possible that the rice *JRL* family genes impart salt tolerance by participating in the ABA signaling pathway. Our RT-qPCR showed that the expression of some salt tolerance genes, including *OsWARKY71*, *OsMYB2*, *OsNAC19*, *OsJAC1*, and *OsCDPK13*, was enhanced by *OsJRL40*. This suggests that the participation of OsJRL40 in salt tolerance may be related to the regulation of the expression of these genes. OsNAC2, a member of the NAC transcription factor family, was strongly induced by ABA and osmotic stresses such as drought and high salt [55]. The expressing *NAC19* gene increased drought and salt tolerance of plants, and the leaf water content and chlorophyll content of these plants were higher than those of the WT [46]. OsWRKY54 regulates the expression of some essential genes related to salt tolerance [47]. The expression of the *OsWRKY71* gene was up-regulated in response to low temperature, salinity, drought, and injury, and its transcript level was the highest under salinity stress [48]. Calmodulin (CAM) subtypes mediate salt-induced Ca^2+^ signaling through the activation of MYB transcriptional activators, leading to salt tolerance in plants [56]. OsGAP1 attenuates salt stress, stimulates a defense response, and binds to phospholipids and the unconventional G protein OsYCHF1 [49]. These results suggest that the involvement of OsJRL40 in salt tolerance is attributable to the regulation of the expression of these genes. 

Overall, this study shows that the *JRL* family gene *OsJRL40* induces salt tolerance in rice by regulating some critical Na^+^/K^+^ transporters, upstream regulatory factors, and stress-related genes.

## 4. Materials and Methods

### 4.1. Plant Material and ABA Treatment

Rice (*Oryza sativa* ssp. *japonica*) cultivar Nipponbare (Nip) was used as the WT in this study. After germination, the seeds were transferred to a normal hydroponic medium, and seedlings were grown to the three-leaf stage over a period of approximately 15 d. Then, the seedlings were grown in 100 mM NaCl for 4 d and subsequently returned to the normal hydroponic medium to recover for 7 d. Finally, the survival rate, fresh weight, and chlorophyll content of seedlings were determined. To observe the salt-tolerant phenotype of rice plants at the reproductive stage, plants at the tillering stage were transferred to 150 mM NaCl-containing soil and grown to maturity before quantifying the agronomic traits. To understand the effect of ABA treatment on rice, seeds and seedlings of WT and mutants (KO1, KO2, OE1, and OE2), they were treated with 0, 2, and 5 μM ABA, respectively. Referring to the previous research and modifying it slightly [57], seeds were inoculated on 1/2 MS medium supplemented with different concentrations of ABA (0, 2, 5 mM) for 5 days. To observe seedling phenotypes, budding seedlings were transferred to 1/2 MS medium supplemented with different concentrations of ABA (0, 2, 5 mM) for about 7 days. Then the phenotype was observed and the root length and shoot length were measured.

### 4.2. Generation of Mutant and Transgenic Lines

The CRISPR/Cas9 method was used to create the *osjrl40* KO mutants. According to the criteria described by Maio and colleagues [58], two 20-bp target sites (1: GGACGTGATCGCCATGGTGG; 2: GGAAAGGGTTGCATCGTCGT) were selected. The CRISPR-Cas9-*OsJRL40* constructs were then transformed into Nip plants via *Agrobacterium*-mediated transformation. The resultant mutant lines were screened for hygromycin resistance and then confirmed by PCR using sequence-specific primers. Two KO mutant lines, KO1 and KO2, were obtained. Primers used for the generation and identification of KO lines are listed in Appendix A.

To construct the *OsJRL40* overexpression plants (OE1 and OE2), the open reading frame of *OsJRL40* was amplified by PCR from rice cDNA using the sequence-specific primer pair 1390-*OsJRL40*-F/R (Appendix A). The purified PCR product was cloned into the pCAMBIA1390-Ubi vector using *Spe*I and *Bam*HI restriction endonucleases. To generate plants expressing the *pOsJRL40: GUS* construct, the *OsJRL40* promoter (2500 bp upstream of ATG) was amplified by PCR from Nip genomic DNA using the primer pair 1301-*OsJRL40*-F/R (Appendix A). The PCR product was cloned into the pCAMBIA1301 binary vector. The resultant plasmid was introduced into *Agrobacterium tumefaciens* strain EHA105, which was then used to transform Nip plants as described previously [59].

### 4.3. Gene Expression Analysis

Total RNA was extracted from whole plants using the TRIzol Reagent (Invitrogen, Waltham, MA, USA) and subjected to reverse transcription to synthesize cDNA. Then, RT-qPCR was performed on the ABI PRISM 7500 RT-qPCR instrument (Applied Biosystems, Waltham, MA, USA) using the Takara quantitative kit RR420A and sequence-specific primers, which were designed with Primer Premier 5 and synthesized by Sangon Biotech (Shanghai, China). The *Osactin* gene was used as an internal reference. 

### 4.4. GUS Staining Assay

GUS staining was performed as described previously [60]. Briefly, the tissues of *ProOsJRL40-GUS* transgenic rice were incubated overnight in a GUS staining solution at 37 °C. Then, chlorophyll was removed using 75% ethanol. Finally, the tissues were examined and photographed using the Olympus SZX7 microscope.

### 4.5. DAB Staining Assay

As described previously [61], Leaves were collected from rice plants before and after the NaCl stress treatment. The leaf samples were incubated in 1 mg mL^−1^ 3,3′-DAB solution (containing 0.05% Tween and 10 mM Na_2_HPO_4_ [pH 3.8]) at 37°C overnight. The stained leaves were discolored with ethanol. Finally, the leaves are transferred to fresh ethanol and imaged.

### 4.6. Electrolyte Leakage Assay

The relative conductivity is as described previously [38]. Rice leaves are washed with deionized water, and the surface water was soaked dry. Next, 25 mL of deionized water was added to the test tube containing the sample and incubated at room temperature for 1 h. The EC of the leachate was measured with a conductivity meter as the initial conductivity (al), and that of the extract (a2) was measured at 0, 2, 4, 6, 8, 10, 12, and 24 h. Finally, the samples were boiled for 15 min. The value of absolute EC (EC after boiling; a3). To determine the cell membrane permeability, relative EC (%) was calculated using the equation below:$$Relative\;EC=\frac{\left(a2-a1\right)}{\left(a3-a1\right)}\times 100$$

### 4.7. Measurement of Antioxidant Enzyme Activities and H_2_O_2_ Content

Plants of all genotypes (WT, KO1, KO2, OE1, and OE2) were grown in normal hydroponic solution for 14 d and then treated with 100 mM NaCl for 2 d. The activities of SOD and POD and the content of H_2_O_2_ were measured using the corresponding assay kits (Jiancheng Bioengineering Institute, Nanjing, China), according to the manufacturer’s instructions.

### 4.8. Measurement of MDA and Chlorophyll Contents

To measure the MDA content, an indicator of the degree of lipid peroxidation [62], fresh rice leaves were ground in trichloroacetic acid on ice. After centrifugation, the supernatant was added to thiobarbituric acid, and the sample was boiled in a water bath for 15 min. After cooling to room temperature, the absorbance of the sample was measured at 450, 532, and 600 nm. To determine the chlorophyll content of leaves, chlorophyll was extracted from leaves (100 mg), and the absorbance of the extract was measured at 663 and 645 nm [38,61].

### 4.9. Subcellular Localization Analysis

To analyze the subcellular localization of OsJRL40, the CDS of *OsJRL40* minus the stop codon was amplified from Nip genomic DNA using gene-specific primers (Appendix A), digested with PstI, and cloned into the PstI-digested pCAMBIA1390-GFP vector. The resultant *35S:OsJRL40-GFP* construct was transiently expressed in rice protoplasts via polyethylene glycol (PEG)-mediated transformation [63]. The GFP signal was detected using the Zeiss LSM 880 microscope [64].

### 4.10. Yeast Complementation Assay

The *OsJRL40* CDS was amplified using gene-specific primers (Appendix A), and then cloned into the pYES2 vector using *Hin*dIII and *Xba*I restriction enzymes. The resultant vector was introduced into BY4741 yeast cells via the LiAC/ssDNA/PEG-mediated transformation method. Yeast growth was observed on a uracil-deficient medium containing 2% galactose (carbon source) and 2%, 5%, or 8% NaCl. 

### 4.11. Ion Content Analysis

Whole plants of all genotypes (WT, KO1, KO2, OE1, and OE2) were ground with a plant grinder, and the powdered samples were passed through a 1 mm sieve. Then, 2 mL of nitric acid–perchloric acid (5:1) mixture was added to 50 mg of the ground sample. The digested sample was filtered, and K^+^ and Na^+^ concentrations were determined by inductively coupled plasma mass spectrometry (ICP-MS) (Agilent 7850).

### 4.12. RNA-Seq Analysis

Whole WT and KO1 plants were sampled before and after the 6 h 100 mM NaCl treatment and sent to Biomarker Technologies for RNA-seq analysis. The RNA-seq data were analyzed as described by Zhang et al. [65]. Then, the DEGs were functionally annotated by performing the GO and KEGG enrichment analyses [66,67].

### 4.13. Statistical Analysis

Figures were constructed using GraphPad Prism5. Data were represented as the mean ± SD of three independent replicates. The mean of three replicates per treatment was calculated using the PASW statistical Software18, and statistically significant differences were detected using one-way analysis of variance (ANOVA) and Student’s *t*-test.

## Figures and Tables

**Figure 1 ijms-24-07441-f001:**
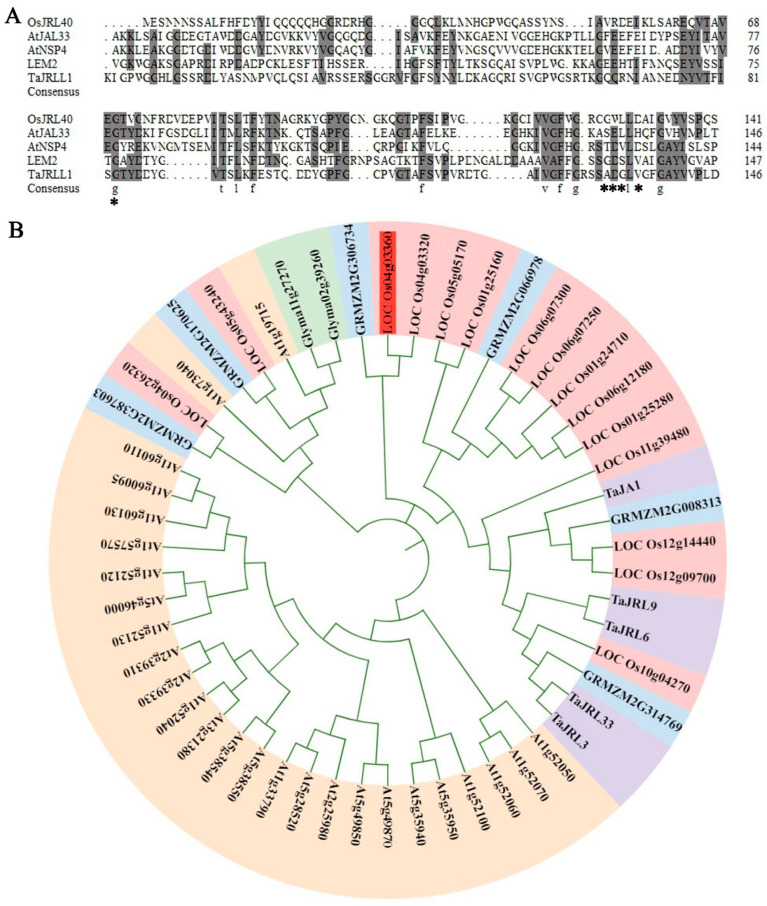
Analysis of OsJRL40 amino acid sequence and phylogenetic relationship. (**A**) Amino acid sequence alignment of OsJRL40, barley LEM2 (AAM18206), TaJRLL1 (ADP37001), AtJRL33 (NP_001030711), and AtNSP4 (NP_188262). Dark shading indicates conserved residues, and invariant residues are marked with lowercase letters. Asterisks indicate mannose-binding sites in the Heltuba protein. ‘*’ represents the mannose-binding site. (**B**) Analysis of the evolutionary relationship between OsJRL40 and JRLs in other species (rice, Arabidopsis, wheat, maize, and soybean). The phylogenetic tree was constructed using the neighbor-joining method in MEGA 7.

**Figure 2 ijms-24-07441-f002:**
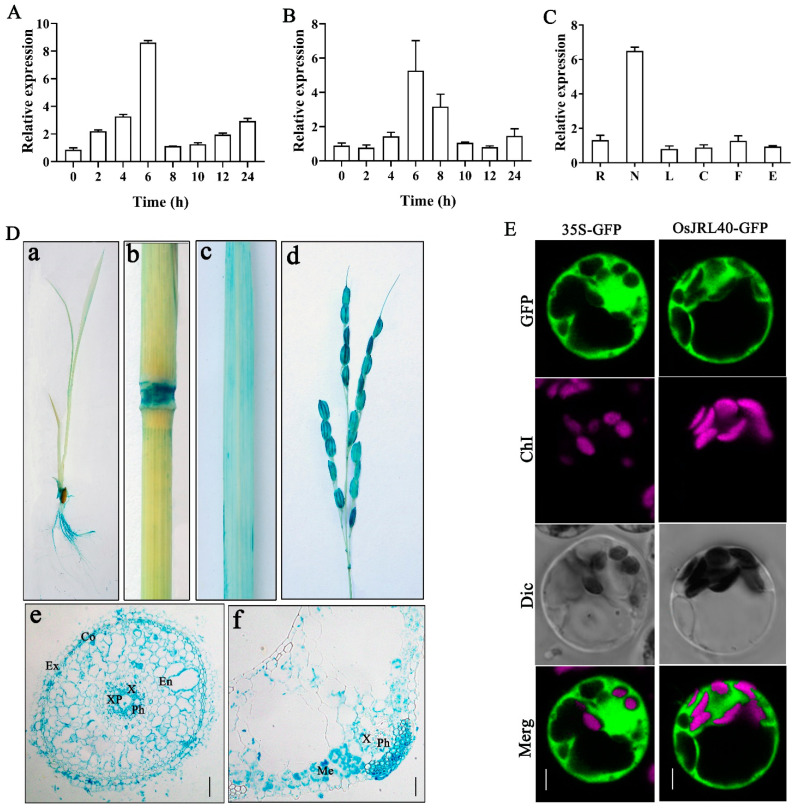
Expression analysis of OsJRL40. (**A**) RT-qPCR analysis of *OsJRL40* in rice seedlings treated with 100 mM NaCl. (**B**) RT-qPCR analysis of *OsJRL40* in rice seedlings treated with 100 uM ABA. (**C**) Transcript levels of *OsJRL40* in different organs of rice plants including the root (R), node (N), leaf (L), culm (C), flower (F), and embryo (E). (**D**) Analysis of GUS staining in transgenic plants expressing the *OsJRL40-Pro: GUS* construct. GUS staining of seedlings (**a**), culms and nodes (**b**), leaves (**c**), panicles (**d**), root cross-sections (**e**), and leaf cross-sections (**f**) are shown. Scale bar = 20 μm. (**E**) Subcellular localization analysis of OsJRL40 in Nip rice by confocal fluorescence microscopy. The red signal represents chlorophyll autofluorescence, and the green fluorescence represents the OsJRL40-GFP fusion protein. Scale bar = 10 μm.

**Figure 3 ijms-24-07441-f003:**
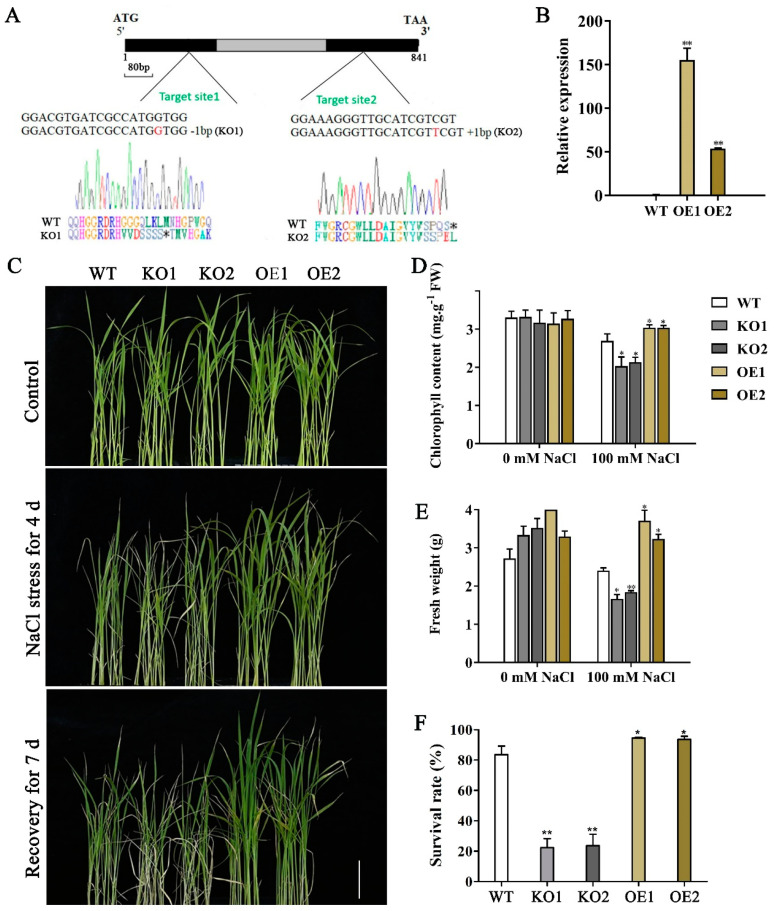
OsJRL40 confers salt stress tolerance in rice. (**A**) Schematic representation of the *OsJRL40* CDS. Black and gray rectangles indicate exons and intron, respectively. The CRISPR/Cas9-targeted sites used for the generation of *osjrl40* KO mutants are indicated. The 1-nt deletion and 1-nt insertion in *OsJRL40* exons in KO1 and KO2 lines, respectively, are also shown. Premature stop codons in the deduced amino acid sequences of the mutant OsJRL40 proteins are indicated with asterisks. (**B**) Relative expression levels of *OsJRL40* in WT, OE1, and OE2 leaves. (**C**) Phenotypic comparison of KO1, KO2, OE1, OE2, and WT seedlings treated with 100 mM NaCl at the three-leaf stage. (**D**–**F**) Chlorophyll content (**D**), fresh weight (**E**), and survival rate (**F**) of KO1, KO2, OE1, OE2, and WT seedlings treated with 100 mM NaCl. Data represent mean ± standard deviation (SD) of three independent replicates. Asterisks indicate statistically significant differences (Student’s *t*-test; * *p* < 0.05, ** *p* < 0.01).

**Figure 4 ijms-24-07441-f004:**
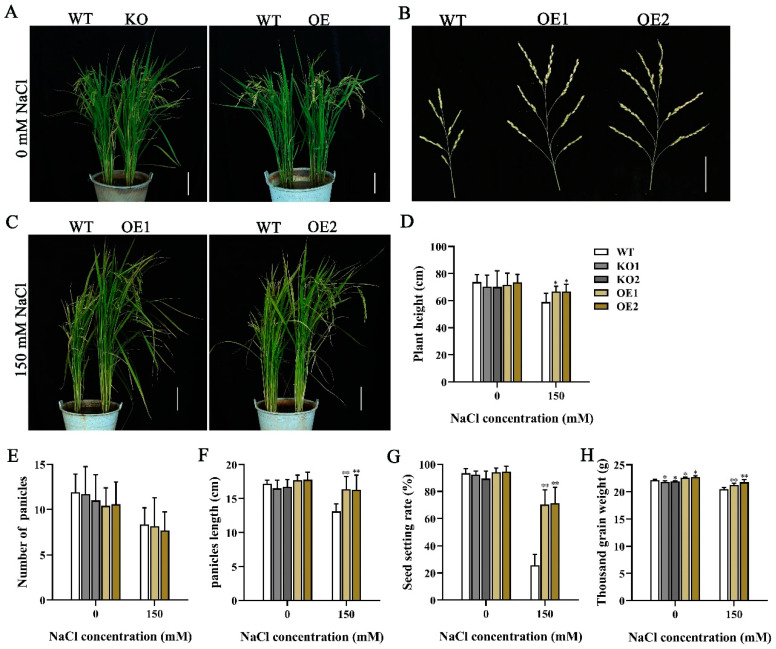
*OsJRL40* enhances salt tolerance of rice plants at the reproductive stage. (**A**–**C**) Phenotypic comparison of WT, KO1, KO2, OE2, and OE2 plants under normal (0 mM NaCl) and salt stress (150 mM NaCl) conditions. Scale bar = 5 cm. (**D**–**H**) Plant height (**D**), effective panicle number (**E**), main panicle length (**F**), seed setting rate (**G**) and thousand grain weight (**H**) of WT, KO1, KO2, OE2, and OE2 plants. Data represent mean ± SD (*n* = 10), and asterisks indicate statistically significant differences (* *p* < 0.05, ** *p* < 0.01; Student’s *t*-test).

**Figure 5 ijms-24-07441-f005:**
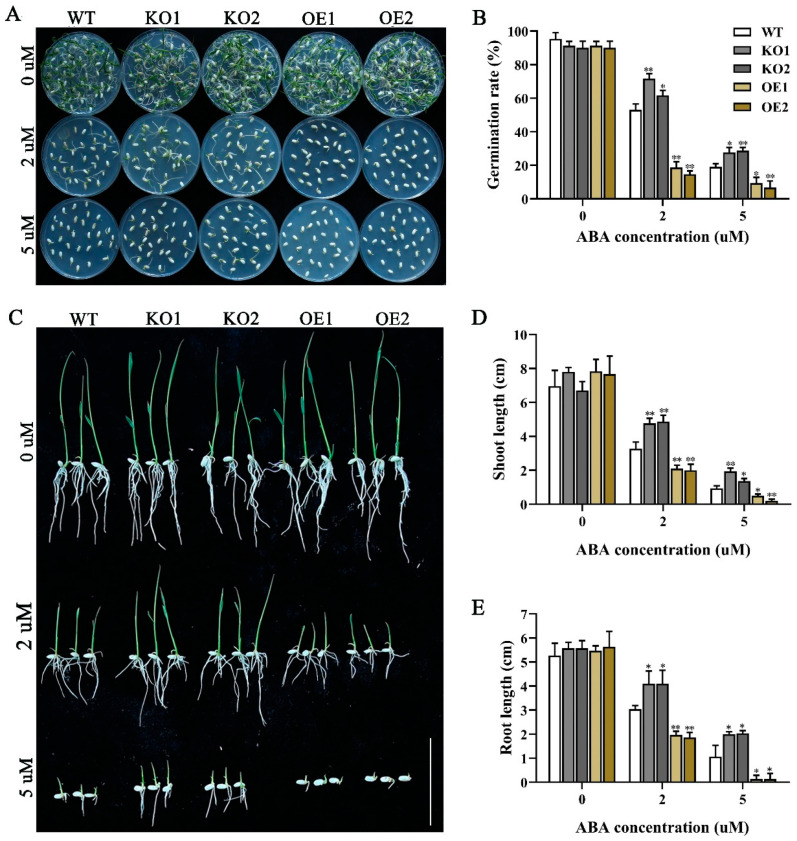
Germination of WT, KO1, KO2, OE1, and OE2 seeds in the ABA treatment. (**A**) Germination of OE1 and OE2 seeds treated with 0, 2, and 5 μM ABA for 5 days. (**B**) Germination rates of WT, KO, and OE seeds on 1/2 MS medium containing 0, 2, and 5 μM ABA. (**C**) Growth of WT, KO, and OE plants treated with 0, 2, and 5 μM ABA after seed germination on 1/2 MS medium for 2 days. Scale bar = 2 cm. (**D**,**E**) Shoot length (**D**) and root length of WT, KO, and OE plants. Data represent the mean ± SD of three independent replicates. Asterisks indicate statistically significant differences (* *p* < 0.05, ** *p* < 0.01).

**Figure 6 ijms-24-07441-f006:**
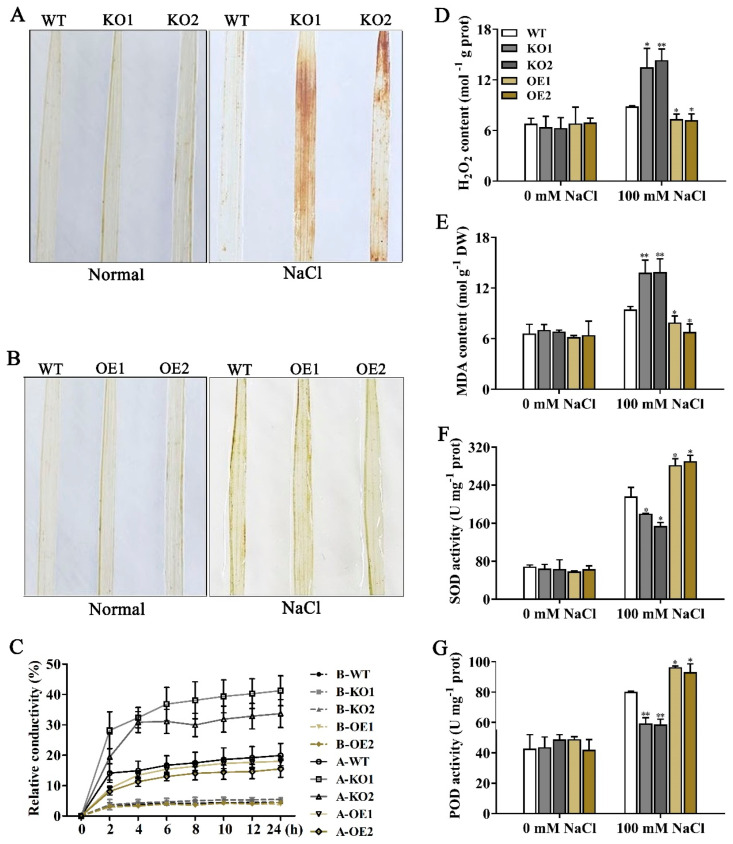
OsJRL40 increased the antioxidant capacity of rice under salt stress. (**A**,**B**) DAB staining of the leaves of WT, KO, and OE seedlings exposed to salt stress for 2 d. (**C**) Percent electrolyte leakage. (**D**) H_2_O_2_ content. (**E**) Malondialdehyde (MDA) content. (**F**) Superoxide dismutase (SOD) activity. (**G**) Peroxidase (POD) activity. In (**C**–**G**), 15-day-old seedlings were treated with 100 mM NaCl for 2 d prior to the sampling of leaves. Data represent the mean ± SD of three independent replicates. Asterisks indicate statistically significant differences (Student’s *t*-test; * *p* < 0.05, ** *p* < 0.01).

**Figure 7 ijms-24-07441-f007:**
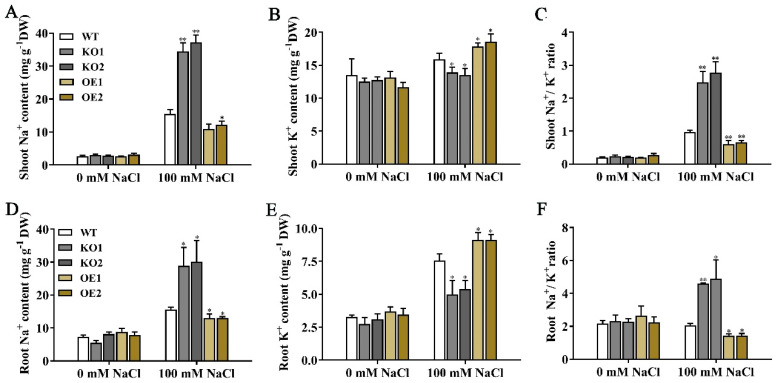
Effects of OsJRL40 on Na^+^ and K^+^ accumulation in rice seedlings under salt stress conditions. Fifteen–day–old seedlings were treated with a hydroponic medium containing 100 mM NaCl for 2 days. (**A**,**B**) Contents of Na^+^ (**A**) and K^+^ (**B**) in shoots. (**C**) Na^+^/K^+^ ratio in shoots. (**D**,**E**) Contents of Na^+^ (**D**) and K^+^ (**E**) in roots. (**F**) Na^+^/K^+^ ratio in roots. Data represent mean ± SD (*n* = 3). Asterisks indicate statistically significant differences (Student’s *t*-test; * *p* < 0.05, ** *p* < 0.01). DW, dry weight.

**Figure 8 ijms-24-07441-f008:**
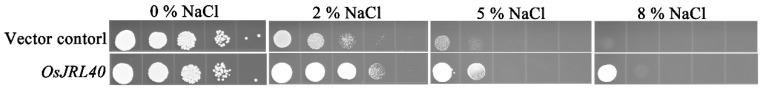
Comparison of the salt tolerance phenotypes of yeast BY4741 cells expressing *OsJRL40* or the empty pYES2 vector control. Ten-fold serial dilutions of yeast cultures were spotted onto plates containing various concentrations of NaCl and incubated at 30 °C. Pictures were taken after 7 days.

**Figure 9 ijms-24-07441-f009:**
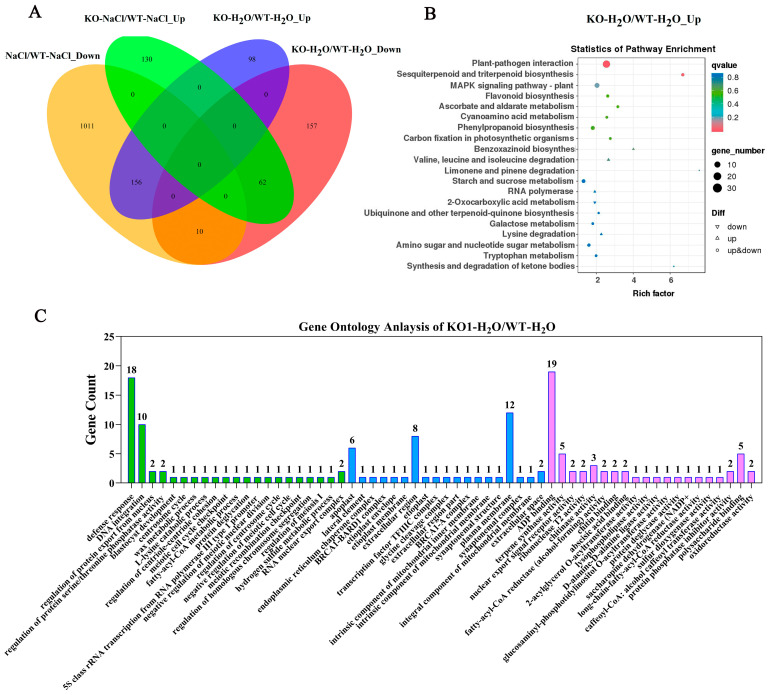
Results of GO and KEGG enrichment analyses of WT and KO1 plants under normal and salt stress conditions. (**A**) Venn diagram showing the overlap of DEGs identified in WT and KO1 seedlings under normal and salt stress conditions. KO-H_2_O/WT-H_2_O-Up/Down: DEGs up-regulated and down-regulated in KO1 plants compared with the WT under normal conditions; KO-NaCl/WT-NaCl-Up/Down: DEGs up-regulated and down-regulated in KO1 plants compared with the WT under salt stress conditions; (**B**) KEGG enrichment analysis of DEGs detected under normal conditions. (**C**) Functional annotation of genes differentially expressed between WT and KO1 plants with normal conditions. These genes were grouped into three categories: biological process (**green columns**), cellular component (**blue columns**), and molecular function (**purple columns**).

**Figure 10 ijms-24-07441-f010:**
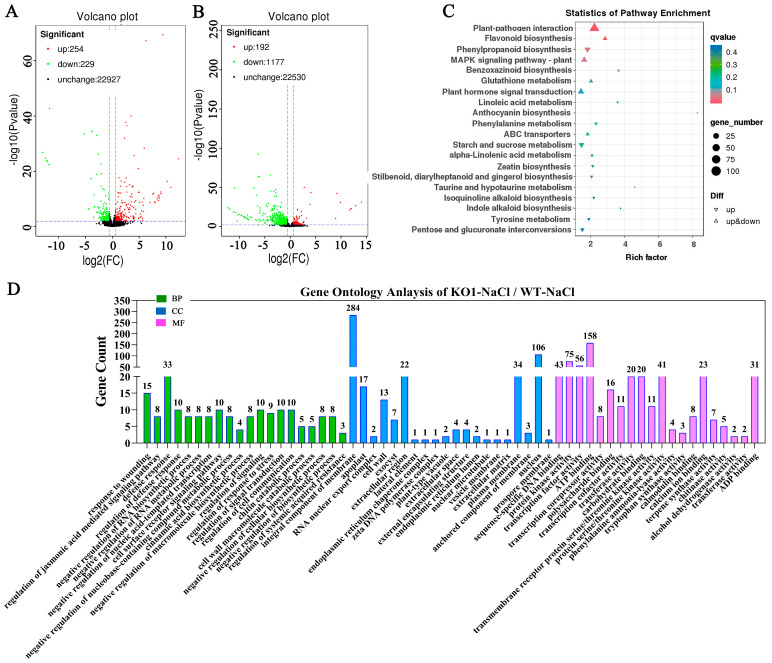
GO and KEGG enrichment analyses of DEGs detected in WT and KO1 plants under salt stress. (**A**,**B**) Volcan plots of DEGs detected under normal conditions (**A**) and salt stress conditions (**B**). (**C**) KEGG enrichment analysis of DEGs detected under salt stress conditions. (**D**) Categorization and functional annotation of DEGs. The different categories were as follows: biological process (**green columns**), cellular component (**blue columns**), and molecular function (**purple columns**).

**Figure 11 ijms-24-07441-f011:**
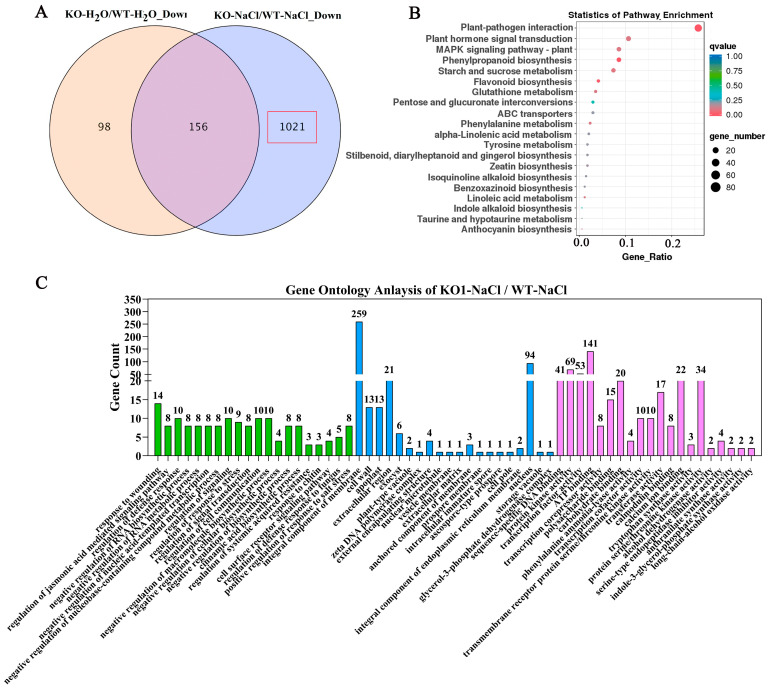
GO and KEGG enrichment analyses of DEGs regulated by *OsJRL40*. (**A**) Venn diagram showing the number of genes up-regulated (fold change > 1.5) in KO1 plants compared with WT plants under normal and salt stress conditions. (**B**) KEGG enrichment results. (**C**) Categorization and functional annotation of DEGs detected in KO1 plants under salt stress. The different categories were as follows: biological process (**green columns**), cellular component (**blue columns**), and molecular function (**purple columns**).

**Figure 12 ijms-24-07441-f012:**
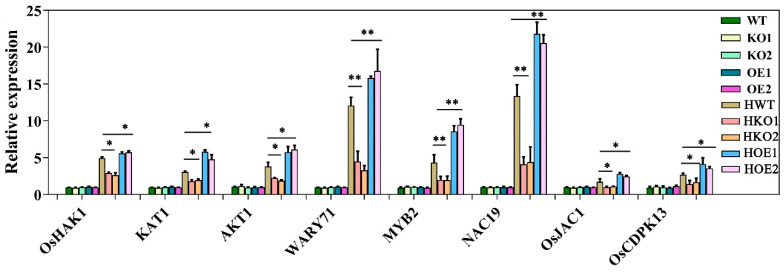
*OsJRL40* regulates the expression of salt stress signaling-related genes under salt stress. WT, KO1, KO2, OE1, and OE2 seedlings at the three-leaf stage were treated with 100 mM NaCl for 8 h. After RNA extraction and reverse transcription, RT-qPCR was performed. *Osactin* was used as the internal control gene. Data represent the mean ± SD of three independent replicates. Asterisks indicate statistically significant differences (* *p* < 0.05, ** *p* < 0.01).

## Data Availability

All data supporting the conclusions of this article are provided within the article and in its additional files and are available upon reasonable request to the corresponding author.

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
