# Peer review of "OsJRL40*, a Jacalin-Related Lectin Gene, Promotes Salt Stress Tolerance in Rice"

_ijms, 2023, doi:10.3390/ijms24087441_

Round 1
Reviewer 1 Report
Introduction,
1. Lines 66-67, please provide more information/details about the research of [32].
2. Line 70, the authors studied the antioxidant related to OsJRL40. In the introduction part, please provide some link between antioxidants, electrolyte leakage, MDA and rice under salinity, or OsJRL.
3. In methods, line 393, the authors treated rice with ABA. So, please provide some link between ABA and rice under salinity, or ABA and OsJRL.
Materials and methods
1. Line 400, seeds, and seedlings were treated with 0, 2, and 5 uM ABA, respectively.
Please provide more details how the authors treated ABA to seeds and seedlings, and what are parameters of data to collect, and when.
2. Method 4.5, please add reference.
3. Method 4.6, please add reference.
4. Method 4.7, what are OE1, OE2? Please explain in the methodology.
5. Method 4.8, please insert reference for chlorophyll extraction and analysis.
6. Method 4.11, please provide instrument detail of ICP-MS, line 477.
7. Method 4.12, please indicate concentration of NaCl. Why is 8-h NaCl treatment? Please explain in the method or insert reference, as 4.7, authors treated rice for 2 d.
Results
1. Line 101, reaching a peak at 8 h, but Figure 2A showed the highest peak at 6 h?.
2. Line 152, 2.4…at the reproductive stage, please provide details in the method section for salt treatment at the reproductive stage. Why the rice was treated with 150 mM NaCl, but 4.1 and 4.7 were treated with 100 mM NaCl, please explain in the manuscript. But the Figure 4 legend line 166 indicated 100 mM NaCl, please check carefully.
3. Please insert mutants in 4.1 for ABA treatments.
Discussion
1. Lines 346-348, please provide references.
Reviewer 2 Report
In the manuscript entitled „OsJRL40, a Jacalin-related Lectin Gene, Promotes Salt Stress Tolerance in Rice” authors show the analysis of the OsJRL40 gene coding for one of the JRL protein in Oryza sativa under stress conditions.
They have prepared loss-of function as well as overexpression mutants of OsJRL40, they have analyzed its promoter activity in different tissues, and they also performed transcriptomic and biochemical analyses.
My comments:
Line 15: "Further molecular analyses showed that OsJRL40 enhances antioxidant enzyme activities and regulates...." - You cannot claim that a gene affects something, because it is the product of the gene that affects a process, in this case this is a protein. A note concerns the whole text.
Line 38: KHT [7,8], NHX [9,10], and HAK - abbreviations should be explained.
Line 113: incorrect reference to fig. 5B
Fig. 2 - incorrect description, what do panels B, C, D and E represents?
Line 405: “via Agrobacterium-mediated transfection.” – for the description of this process “transformation” should be used.
Line 462: “digested with XbaI, and cloned into the XbaI-digested pCAMBIA1390 vector” – there is no sequence coding for GFP in pCAMBIA1390 vector, as well as XbaI restriction site suitable for cloning. Also pCAMBIA1390-Ubi vector should be better described.
Reviewer 3 Report
Much of what authors are reporting has already been reported previously except that "OsJRL40" is a newly reported gene with its function elucidation.
https://www.ncbi.nlm.nih.gov/pmc/articles/PMC6921557/
https://onlinelibrary.wiley.com/doi/10.1111/plb.12514
I urge the authors to clarify a bit more about OsJRL40 in the introduction.
